# The effect of atom losses on the distribution of rapidities in the one-dimensional Bose gas

Isabelle Bouchoule[1], Benjamin Doyon[2] and Jérôme Dubail[3]

**1** Laboratoire Charles Fabry, Institut d'Optique, CNRS, Université Paris Sud 11, 2 Avenue Augustin Fresnel, 91127 Palaiseau Cedex, France
**2** Department of Mathematics, King's College London, Strand WC2R 2LS, UK
**3** Université de Lorraine, CNRS, LPCT, F-54000 Nancy, France

## Abstract

We theoretically investigate the effect of atom losses in the one-dimensional (1D) Bose gas with repulsive contact interactions, a famous quantum integrable system also known as the Lieb-Liniger gas. The generic case of $K$-body losses ($K = 1, 2, 3, \dots$) is considered. We assume that the loss rate is much smaller than the rate of intrinsic relaxation of the system, so that at any time the state of the system is captured by its rapidity distribution (or, equivalently, by a Generalized Gibbs Ensemble). We give the equation governing the time evolution of the rapidity distribution and we propose a general numerical procedure to solve it. In the asymptotic regimes of vanishing repulsion – where the gas behaves like an ideal Bose gas – and hard-core repulsion – where the gas is mapped to a non-interacting Fermi gas –, we derive analytic formulas. In the latter case, our analytic result shows that losses affect the rapidity distribution in a non-trivial way, the time derivative of the rapidity distribution being both non-linear and non-local in rapidity space.

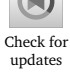

# 1  Introduction

Nowadays, trapped ultracold atoms offer versatile platforms for the study of isolated quantum many-body dynamics. The system is never perfectly isolated though, and it is always weakly coupled to its environment. The main effect breaking unitary evolution is the atom losses. Although of primary interest to understand the limitations of the simulation of quantum many-body physics, a complete theoretical description of losses is still lacking. Different loss processes occur in cold atom experiments: one-body losses might be non negligible; two-body losses due to inelastic two-body collisions are sometimes present [1] or engineered [2]; three-body losses, where a deeply bound diatomic molecule is formed, are always present and are usually dominant [3,4]. All those $K$-body loss processes – where $K$ is the number of atoms involved, and lost, in each loss event – are local and their effect on the mean atomic density $n$ reads $dn/dt = -Gn^K K g_K$. Here $g_K$ is the normalized zero-distance $K$-body correlation function and $G$, which has units of length$^{d(K-1)}$.time$^{-1}$ for a gas of dimension $d$, is the constant quantifying the loss process. However, the many-body state at time $t$ is not characterized solely by its atom density $n$. In particular, the above equation is not a closed equation for $n$, because the determination of the correlation function $g_K$ itself requires knowledge of the many-body state. For chaotic systems, an important simplification occurs if the rate $Gn^{K-1}$ is much smaller than the relaxation time of the system. Then, as far as local observables are concerned, the system is described at any time by a thermal state which is entirely determined by the mean atomic density $n$ and mean energy density $e$. The time derivative of the mean atom density $n$, a local observable, can be computed once $g_K$ is calculated for the thermal state [5]. Computing the time evolution of $e$ is more difficult. This can be done for a system where interactions are weak, since the mean energy of a lost atom is simply $e/n$. The time-evolution of the system can then be computed [6,7].

However, such an analysis is not valid for one-dimensional (1D) bosons with point-like repulsive interactions, also known as the Lieb-Liniger gas [8]. This quantum gas is a famous integrable system [9, 10], and in the past fifteen years it has been established both experimentally and theoretically that it does not thermalize [11, 12], see the reviews in the special issue [13]. Relaxation is still meaningful but, owing to the infinite number of conserved quantities, the system after relaxation is described not only by two quantities $n$ and $e$, but by a whole function, known as the rapidity distribution [14–17]. Several early works on atom losses – predating the ones on the absence of thermalization – have focused on the calculation of $g_3$ in the ground state of the Lieb-Liniger gas [18,19] and $g_2$ in thermal states [20–23]. These results were soon extended to excited states of the gas [24,25], culminating in general expressions for $g_K$ valid for arbitrary rapidity distributions [26,27]. However, these results are not sufficient to fully describe the evolution of the system under atom losses. Recent attempts in that direction have concentrated on the quasicondensate regime. This regime is characterized by weak correlations between atoms, with $g_K \simeq 1$, and is well modeled using a Bogoliubov approximation where the system is described by a collection of independent collective modes. The evolution of the energy in each collective mode under the loss process was computed in

Refs. [28, 29] and, in particular, it was shown that the system evolves towards a non-thermal state [30]. Predictions concerning phonons were verified experimentally [31–33].

In this paper, we revisit the problem of atom losses in the Lieb-Liniger gas. We assume that the loss process is slow compared to the intrinsic dynamics, in such a way that the system at any time is described, locally, by a rapidity distribution $\rho(k)$, or equivalently by a Generalized Gibbs Ensemble (GGE). We give a complete description of the effects of losses by computing the evolution of the rapidity distribution. We devise a numerical procedure valid for any initial state. In the two asymptotic cases where the gas lies in the ideal Bose gas and Tonks-Girardeau [34] (hard core) regimes, we obtain analytical expressions for the evolution of the rapidity distribution.

The rapidity distribution is a key notion for this paper. Its basic definition does not require integrability, and is based on the notion of asymptotic states in scattering theory. Imagine that a homogeneous Lieb-Liniger gas is confined in a flat box potential of length $L$ and that the box potential is suddenly released so that the gas expands freely in 1D. The rapidity distribution $\rho(k)$ of the original state is simply the density, per unit asymptotic velocity $k$ and per unit length of the original box, of asymptotic particles obtained after an infinite time of this 1D expansion of the gas. What is special about integrable models is that, by elastic and factorised scattering [35, 36], the rapidity distribution $\rho(k)$ is conserved by the dynamics. Thus the rapidity distribution is a good parametrisation of any state after relaxation. Crucially, by this definition, $\rho(k)$ is also a measurable quantity. The above thought experiment typical of scattering theory is nothing but a 1D expansion [37–41]. After a sufficiently long expansion time $t_{1D}$ the atoms are propagating freely and their velocities are nothing but the rapidities. Thus, measuring the velocity distribution at $t_{1D}$ amounts to measuring the rapidity distribution, as has been very recently done for a Lieb-Liniger gas in the hard core regime [42]. Note that the atoms' velocity distribution is not conserved by the dynamics and the initial velocity distribution is different from the rapidity distribution. The velocity distribution at $t_{1D}$ can be measured by time-of-flight, for instance letting the gas expand further in 1D and measuring the density profile, homothetic to that of velocities. The rapidity distribution is an object of central experimental relevance in the investigation of out-of-equilibrium 1D gases, and understanding how it is affected by atom losses is therefore of paramount importance. This is what we do in this paper.

## 2 Pinpointing the problem

In principle, one would like to describe the gas by a density matrix $\hat{\rho}$ – not to be confused with the distribution of rapidities $\rho(k)$ – evolving according to the Markovian Lindblad equation

$$\frac{d\hat{\rho}}{dt} = -i[H, \hat{\rho}] + G \int_0^L \left( \Psi^K \hat{\rho} \Psi^{+K} - \frac{1}{2} \{ \Psi^{+K} \Psi^K, \hat{\rho} \} \right) dx. \tag{1}$$

Here $\Psi = \Psi(x)$ is the bosonic atom annihilation operator, and $H = \int_0^L \Psi^+ \left( -\partial_x^2/2 + (g/2)\Psi^+\Psi \right) \Psi \, dx$ is the Hamiltonian of the Lieb-Liniger gas. We set $\hbar = m = 1$. The dimensional parameter $G$ is the same as in the introduction; it quantifies the loss process.

As the size of the density matrix $\hat{\rho}$ is exponential in the number of atoms $N$, the Lindblad equation is not tractable in physically relevant setups where $N \sim 10^2 - 10^5$. Fortunately, in the asymptotic limit of small $G$, where the dynamics generated by losses is slow, the complexity of the problem is dramatically reduced.

The key notion that permits simplification of the problem is the notion of relaxation alluded to above. Recall that an isolated Lieb-Liniger gas relaxes at long times, in the sense that

averages of local observables approach asymptotic values. For lengths of systems typically found in experiments, $\tau$ is much smaller than the Poincaré recurrence time, thus relaxation is a good approximation. As argued above, the asymptotic values of local observables are all determined by a single intensive function, the rapidity distribution $\rho(k)$. How do we evaluate the averages of local observables in terms of $\rho(k)$? For this purpose, we use the techniques of integrability. The eigenstates of Bethe Ansatz form naturally encode the asymptotic velocities of the scattering states. An eigenstate is parametrised by a set of rapidities $|\{\lambda_i\}\rangle$, and its associated rapidity distribution is simply $\rho_{\{\lambda_i\}}(k) = L^{-1}\sum_i \delta(k - \lambda_i)$: $L\rho(k)dk$ is the number of rapidities in the interval $[k, k+dk]$. At large $L$, the rapidities form a continuum. To compute mean values of local quantities, we consider a subsystem of length $\ell$ where $\ell$ is much larger than the correlation lengths of the system but much smaller than $L$. The rest of the system acts as a reservoir of rapidities, and the subsystem relaxes to a GGE. Up to corrections of order $1/\ell$, the reduced density matrix of this subsystem is diagonal in the basis of its Bethe states and takes the form $\hat{\rho}_{\text{GGE}} = \sum_{\{\lambda_i\}} p(\{\lambda_i\})|\{\lambda_i\}\rangle\langle\{\lambda_i\}|$. The explicit form of the distribution $p(\{\lambda_i\})$, given $\rho(k)$, is determined by entropy maximisation with the constraint that, for all $k$, $\sum_{\{\lambda_i\}} p(\{\lambda_i\})\rho_{\{\lambda_i\}}(k) = \rho(k)$. This is a simple generalization [17] of the thermodynamics calculations done in [14].

We assume that the loss process occurs on times much longer than the intrinsic relaxation time of the system, $\tau$. More precisely, the typical rate $Gn^{K-1}$ is assumed to be much smaller than $1/\tau$. Then one can assume that, at any time $t$, the system has relaxed with respect to its Hamiltonian $H$. Thus, according to the above considerations, the system at any time $t$ is completely determined by its rapidity distribution $\rho(k)$. The reduction of the complexity of the problem stems from having replaced the full density matrix $\hat{\rho}$ by the one-dimensional function $\rho(k)$. To lowest order in $Gn^{K-1}\tau$, the Lindblad equation leads to

$$\frac{d}{dt}\rho(k) = -Gn^{K-1}F[\rho](k). \tag{2}$$

Since $\int \rho(k)dk = n$ and $dn/dt = -GK\langle\Psi^{+K}\Psi^K\rangle$, we see that $F$ is related to the $K$-body local correlation $g_K = \langle\Psi^{+K}\Psi^K\rangle/n^K$ as $\int F[\rho](k)dk = Kn\, g_K$. Here $\langle\Psi^{+K}\Psi^K\rangle$ denotes the local correlation function $\langle\Psi^{+K}(x)\Psi^K(x)\rangle$, which does not depend on $x$.

The key problem is to determine the functional $F$, which is the main goal of this paper. The evolution of rapidity distributions, or GGEs, under general Lindbladian dynamics has been studied in [43–46], however the particular problem of losses has not been addressed yet.

## 3 The functional $F$ as an expectation value of a local observable

In order to determine the functional $F$, the Lindblad equation (1) is translated into an evolution equation for averages of local quantities $q(x)$, obtained by inserting Eq. (1) into $d\langle q(x)\rangle/dt = \text{Tr}(q(x)d\hat{\rho}/dt)$. Eq. (1) is translational invariant so we can assume that $\hat{\rho}$ also is, and we omit the variable $x$ in $\langle q(x)\rangle$. In Eq. (1), the contribution of the Hamiltonian term can be written, using cyclic invariance of the trace, as the mean value of the commutator $[q, H]$. The latter is a local quantity since $q$ is local. [Although $H$ is not a local operator, it is an integral of local operators so its commutator with the local operator $q$ is local.] One can therefore use the GGE density matrix $\hat{\rho}_{\text{GGE}}$ to represent its average, and we then find that the contribution of this term vanishes since $[H, \hat{\rho}_{\text{GGE}}] = 0$. Thus only the non-hermitian term contributes to $d\langle q\rangle/dt$. Using translational invariance of $\hat{\rho}$, the integral over $x$ can be recast into a form which involes the total charge $Q = \int_0^L q(x)dx$ and we obtain

$$\frac{d\langle q\rangle}{dt} = \frac{G}{2}\Big(\langle\Psi^{+K}(0)[Q, \Psi^K(0)]\rangle_{[\rho]} + \langle[\Psi^{+K}(0), Q]\Psi^K(0)\rangle_{[\rho]}\Big). \tag{3}$$

The notation $\langle\ldots\rangle_{[\rho]}$ means that the expectation values are computed in the GGE corresponding to $\rho(k)$, which is justified since the operators inside the brackets are local operators, $Q$ appearing only inside a commmutator with a local operator. For pedagogical purposes, we rederive (3) using a toy model where lost atoms are absorbed by an environment of oscillators, in Appendix B.

The evolution of the distribution of rapidities $\rho(k)$ is obtained by choosing $q_k$ such that the total charges $Q_k$ are the conserved quantities of eigenvalues $Q_k|\{\lambda_i\}\rangle = \sum_i \delta_\sigma(k-\lambda_i)|\{\lambda_i\}\rangle$. Here $\delta_\sigma(\lambda)$ is any approximation of the delta function with a rapidity spread of order $\sigma$ (for instance a Gaussian of width $\sigma$), where we choose $\sigma \gg 1/L$; as a consequence, the density $q_k$, of extent $\sigma^{-1}$ in position space [47], is local. Choosing $\sigma \ll \Delta k$, where $\Delta k$ is the scale over which $\rho(k)$ varie, one has $\langle q_k \rangle \simeq \rho(k)$ and (3) gives

$$F[\rho](k) = -n^{1-K}\langle\Psi^{+K}(0)[Q_k,\Psi^K(0)]\rangle_{[\rho]}, \tag{4}$$

where we used the fact that $\hat{\rho}_{\text{GGE}}$ commutes with the conserved quantity $Q_k$. The formulation (2) of the problem, and the definition (4) of the functional $F$, is the first main result of our paper. In the following, we use Eq. (4) to compute $F$.

## 4 General case: numerical summation over Bethe states

To evaluate Eq. (4), one must be able to calculate expectation values $\langle\ldots\rangle_{[\rho]}$. Analytically, this is a very hard problem, and at present there exists no general method to solve it – at least, not for arbitrary repulsion strength $g$ –. Therefore, in this paragraph, we turn to numerics and design a general numerical method to evaluate $F$.

Eq. (4) can be evaluated numerically in finite size $\ell$ by computing a double sum over Bethe states. The first sum comes from the expectation value $\langle\ldots\rangle_{[\rho]}$, taken with respect to the GGE parameterized by the rapidity distribution $\rho(k)$. This is a diagonal density matrix in the basis of Bethe states $|\{\lambda_i\}\rangle$, with entries $p(\{\lambda_i\})$,

$$p(\{\lambda_i\}) = \frac{1}{Z}\exp\left[-\sum_i W[\rho](\lambda_i)\right]. \tag{5}$$

$Z$ is a normalization factor such that $\sum_{\{\lambda_i\}} p(\{\lambda_i\}) = 1$, and the weight $W[\rho](\lambda)$ is related to $\rho$ by the (generalized) thermodynamics Bethe Ansatz equation of Yang and Yang [14] – with the differential scattering phase $K(\lambda-\lambda') = 2g/(g^2+(\lambda-\lambda')^2)$ of the Lieb-Liniger model [8] –

$$
\begin{aligned}
W[\rho](\lambda) &= \log(\rho_s(\lambda)/\rho(\lambda)-1) - \int\frac{d\lambda'}{2\pi}K(\lambda-\lambda')\log(1-\rho(\lambda')/\rho_s(\lambda')), \\
\rho_s(\lambda) &= \frac{1}{2\pi} + \int d\lambda' K(\lambda-\lambda')\rho(\lambda').
\end{aligned}
\tag{6}
$$

The second sum comes from inserting a set of intermediate states, $1 = \sum_{\{\mu_j\}}\left|\{\mu_j\}\right\rangle\left\langle\{\mu_j\}\right|$, between the observables $\Psi^{+K}(0)$ and $[Q_k,\Psi^K(0)]$ in Eq. (4). The commutator action is evaluated by using the eigenvalue equation for $Q_k$, and we obtain

$$F[\rho](k) = n^{1-K}\sum_{\substack{|\{\lambda_i\}\rangle \\ |\{\mu_j\}\rangle}} p(\{\lambda_i\})|\langle\{\mu_j\}|\Psi(0)^K|\{\lambda_i\}\rangle|^2 \left(\sum_i\delta_\sigma(k-\lambda_i) - \sum_j\delta_\sigma(k-\mu_j)\right). \tag{7}$$

This expression of $F$ in terms of the form factors $\langle\{\mu_j\}|\Psi(0)^K|\{\lambda_i\}\rangle$ of the Bethe states is the second main result of our paper. The physical meaning of this equation is clear. If the initial state of the system is $|\{\lambda_i\}\rangle$, the probability to have a loss event during the time interval $dt$ and that the system at $t+dt$ is found in the state $|\{\mu_j\}\rangle$ is $\ell G dt|\langle\{\mu_j\}|\Psi(0)^K|\{\lambda_i\}\rangle|^2$.

In such a case, the final value of $q_k$ is $q_k = (1/\ell)\sum_i \delta_\sigma(k - \mu_i)$. The probability for the system to stay in the initial state, and thus that $q_k$ stays equal to $(1/\ell)\sum_i \delta_\sigma(k - \lambda_i)$, is $(1 - \ell G dt \sum_{|\{\mu_j\}\rangle} |\langle\{\mu_j\}|\Psi(0)^K|\{\lambda_i\}\rangle|^2) = (1 - \ell G dt \langle\Psi^{+K}\Psi^K\rangle)$. Computing $\langle q_k\rangle$ summing over the different cases detailed above, and using $\rho(k) \simeq q_k$, we arrive at Eq. (7).

Introducing the conditional probability $p(\{\mu_j\}|\{\lambda_i\}) = |\langle\{\mu_j\}|\Psi(0)^K|\{\lambda_i\}\rangle|^2/\langle\{\lambda_i\}|\Psi^{+K}\Psi^K|\{\lambda_i\}\rangle$, as well as the $K$-body correlation in a given Bethe state $g_K(\{\lambda_j\}) = \langle\{\lambda_j\}|\Psi^{+K}\Psi^K|\{\lambda_j\}\rangle/n^K$, we can rewrite Eq. (7) as

$$F[\rho](k) = n \sum_{\substack{|\{\lambda_i\}\rangle \\ |\{\mu_i\}\rangle}} p(\{\lambda_i\}) p(\{\mu_j\}|\{\lambda_i\}) \, g_K(\{\lambda_i\}) \left( \sum_i \delta_\sigma(k - \lambda_i) - \sum_j \delta_\sigma(k - \mu_j) \right). \tag{8}$$

This enables us to evaluate $F$ numerically, by measuring the expectation value of $g_K(\{\lambda_i\})(\sum_i \delta_\sigma(k - \lambda_i) - \sum_j \delta_\sigma(k - \mu_j))$ with respect to the probability distribution $p(\{\lambda_i\}) p(\{\mu_j\}|\{\lambda_i\})$. To do this, we sample pairs of Bethe states $|\{\lambda_i\}\rangle, |\{\mu_j\}\rangle$, using two Markov chains which have equilibrium distributions $p(\{\lambda_i\})$ and $p(\{\mu_j\}|\{\lambda_i\})$ respectively. Our two Markov chains are constructed using a Metropolis-Hastings algorithm, by implementing moves of Bethe integers and solving the Bethe equations [8–10] with a Newton-Raphson method to find the associated configurations of rapidities $\{\lambda_i\}$ and $\{\mu_j\}$ (see Refs. [48–50] for similar numerical summations over Bethe states, and especially Refs. [51,52] for similar samplings of GGEs). Crucially for our method, exact analytical formulas are available for the form factors of $\Psi(0)^K$ thanks to recent work by Piroli and Calabrese [53], and for $g_K(\{\lambda_j\})$ thanks to work by Pozsgay [26]. Our numerical procedure heavily relies on these exact formulas.

The procedure is computationally costly, however at present we do not know of any realistic alternative to evaluate the functional $F$ numerically. In Fig. (1), we show the results obtained with this method, for the rapidity distribution $\rho(k)$ of a thermal state at $T = 0.2n^2$ and $g/n = 1$, for $K = 1, 2, 3$. The function $\delta_\sigma(k)$ approximating the delta function in rapidity space is a Gaussian of width $\sigma = 0.06n$. To obtain these results we work with $N \simeq 30$ particles in average (the number of particles is let to fluctuate around some fixed mean value in our code), and sum over $10^5$ independent pairs of states $|\{\lambda_i\}\rangle, |\{\mu_j\}\rangle$. We have checked that the two Markov chains are long enough so that the pairs are truly independent, and that the results do not significantly change as we increase $N$ (see Fig. 1). In total, the computation shown in Fig. 1 takes about 10 hours on a laptop. Notice that, since it is essentially a Monte Carlo integration method, our procedure can be trivially parallelized. This could be important for practical purposes.

Finally, we would like to stress that, in principle, for a sufficiently large system size $\ell$, a single Bethe state would be sufficient to evaluate the expectation value (4): according to the idea of typicality of eigenstates –sometimes referred to as Generalized Eigenstate Thermalisation Hypothesis–, the expectation values of local observables in Bethe Ansatz states are smooth functionals of the rapidity distribution and do not depend on details of the specific eigenstate considered (see e.g. Refs. [54–56]). Thus, instead of a double sum, a single sum needs to be evaluated, in principle (see for instance Ref. [57] where this property is exploited to numerically evaluate the dynamical density-density correlation). However, because our observable is the rapidity distribution itself, we find that this idea does not work in practice: the discrete nature of the rapidities induces a rugosity of their distribution at finite $\ell$, and huge system sizes $\ell$, hardly tractable numerically, would be necessary to mitigate this effect. We find that, for our purposes, the above method works and leads to reliable numerical results, while using a single Bethe state as in Ref. [57] doesn't.

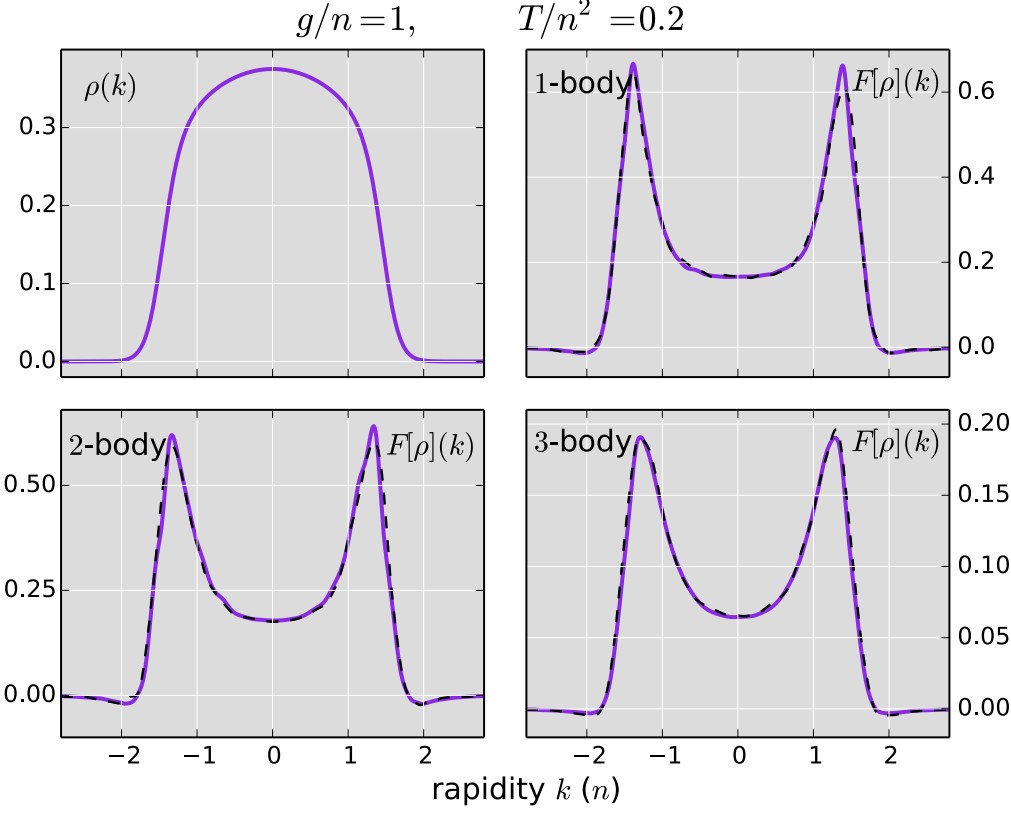

Figure 1: The rapidity distribution $\rho(k)$ of the thermal state at temperature $T = 0.2n^2$, with dimensionless repulsion strength $g/n = 1$, is shown in the top left panel. The other three panels show the corresponding functional $F[\rho](k)$ computed numerically using Eq. (8), for $K = 1, 2, 3$ respectively. The purple line is obtained by summing over $10^5$ pairs of Bethe states with an average number of atoms $N \simeq 30$, and a smoothened density corresponding to a Gaussian convolution with width $\sigma = 0.06n$. The black dashed line is obtained with $N \simeq 60$; it shows that the results are converged as a function of system size/atom number.

## 5 Ideal Bose gas limit

When the typical energy per atom $E$ is much larger than both the scattering energy $g^2(= mg^2/\hbar^2)$ and the interaction energy $gn$, interactions play a negligible role and the gas is well described by an ideal Bose gas. $E \gg g^2$ ensures that a collision event between two atoms leads to negligible reflexion [58], while $E \gg gn$ ensures that the bosons are far from the quasicondensate regime. The rapidity distribution is then simply the momentum distribution of the ideal Bose gas: because the gas is free, this momentum distribution is what would be measured by a 1D time of flight. More precisely, defining the canonical bosonic operators $\Psi_k = \int_0^\ell e^{-ikx}\Psi(x)dx/\sqrt{\ell}$, which annihilate an atom of momentum $k$ (with values in $2\pi\mathbb{Z}/\ell$), the Hamiltonian reduces to $H = \sum_k (k^2/2)\Psi_k^\dagger\Psi_k$ and the rapidity distribution is $\rho(k) = \langle\Psi_k^+\Psi_k\rangle/(2\pi)$. In order to compute the evolution of the rapidity distribution due to losses, we use again equation (4). In this expression, the delta-function approximation for the operator $Q_k$ may be chosen simply as $\delta_\sigma(k) = \ell\delta_{k,0}/(2\pi)$. Therefore, $Q_k = \ell\Psi_k^+\Psi_k/(2\pi)$. The density matrix is $\hat{\rho}_{\text{GGE}} = \prod_k \hat{\rho}_k$ where $\hat{\rho}_k$ is gaussian, such that one can use Wick's theorem.

The commutator in (4) is immediately obtained as

$$[Q_k, \Psi(0)^K] = -K \frac{\sqrt{\ell}}{2\pi} \Psi_k \Psi(0)^{K-1},$$
(9)

and the expression on the right-hand side of (4) is evaluated using Wick's theorem, with contractions $\langle \Psi^+(0)\Psi(0) \rangle_{[\rho]} = n$ and $\frac{\sqrt{\ell}}{2\pi}\langle \Psi^+(0)\Psi_k \rangle_{[\rho]} = \rho(k)$. The result is therefore

$$F[\rho](k) = KK!\rho(k).$$
(10)

We see that the rapidity distribution keeps the same form, being simply rescaled in amplitude as time goes on. Now suppose that the initial state is thermal at temperature $T$ and chemical potential $\mu$. In the ideal Bose gas regime ($\mu < 0$, $|\mu| \gg g^{2/3}T^{2/3}$ and $T \gg g^2$ [21]), $\rho(k)$ is therefore close to the Bose-Einstein distribution $\frac{1}{2\pi}/(e^{(k^2/2-\mu)/T}-1)$. However, a Bose-Einstein distribution rescaled in amplitude –*i.e.* multiplied by an overall constant factor– is no longer a Bose-Einstein distribution [1], unless the gas is in the classical regime where $T \ll |\mu|$ (which corresponds to $n \ll \sqrt{T}$ [2]). Thus the property of the state being thermal is not something that is preserved under losses: the system's state become non-thermal. This is expected to be a generic property of integrable systems, as argued for general Lindbladian evolution in [43–46].

In Fig. 2, we display our numerical results for the thermal state at temperature $T = 5n^2$ and repulsion strength $g = 0.1n$. This is close to the ideal Bose gas regime, although deviations of $\rho(k)$ from the Bose-Einstein distribution are clearly visible at small $k$. We compute $F$ numerically using Eq. (7), and we find that the results are in good agreement with formula (10). On the technical side, the numerical evaluation of the double sum (7) is more difficult than in the regime shown in Fig. 1, because the auto-correlation time of our Markov chain is much longer. We thus work with smaller samples of pairs of Bethe states ($10^4$ pairs for $K = 1$ and $2.10^3$ for $K = 2$), which explains the fluctuations visible in Fig. 2 (especially for $K = 2$).

## 6 Tonks-Giradeau limit

The hard-core regime is obtained when the typical energy per atom fulfills $E \ll g^2$. The probability to find more than one atom at a given position is vanishing in this regime, so that only the case $K = 1$ is relevant. We restrict to this case here. It is well-known that, in this regime, the Lieb-Liniger gas is mapped to non-interacting fermions by the Jordan-Wigner transformation $c(x) = (-1)^{N_{[0,x]}}\Psi(x)$, where $N_{[0,x]}$ is the number of atoms in the interval $[0,x]$. This transformation ensures the canonical anticommutation rules, $\{c^+(x), c(y)\} = \delta(x-y)$. Two sets of mode operators can be defined, $c_\lambda = \frac{1}{\sqrt{\ell}}\int e^{-i\lambda x}c(x)dx$ for either $\lambda \in 2\pi\mathbb{Z}/\ell \equiv \mathbb{Z}_p$, or $\lambda \in 2\pi(\mathbb{Z}+\frac{1}{2})/\ell \equiv \mathbb{Z}_{ap}$. With the vacuum $|0\rangle$, both sets of states $\{c_\lambda^+|0\rangle\}$ and $\{c_\mu^+|0\rangle\}$, for $\lambda \in \mathbb{Z}_p$ and $\mu \in \mathbb{Z}_{ap}$, form a basis of the one-particle Hilbert space $L^2(\ell)$. They are related to each other as

$$c_\mu = \frac{2i}{\ell}\sum_{\lambda \in \mathbb{Z}_0}\frac{1}{\lambda - \mu}c_\lambda,$$
(11)

---

[1]Consider an ideal Bose gas in a thermal state. Then the momentum distribution $\rho(k)$ has Gaussian large-$k$ wings, with expansion $\rho(k) = 1/(2\pi)\left[e^{\mu/T}e^{-k^2/(2T)} + e^{2\mu/T}e^{-k^2/T} + \dots\right]$. Clearly, the rescaled distribution $\alpha\rho(k)$ (for $\alpha > 0$, $\alpha \neq 1$ constant) has Gaussian wings which correspond to that of a gaz at temperature $T$ and chemical potential $\mu + T\ln(\alpha)$. But at the next order of the expansion, this change of chemical potential does not reproduce the scaling. Thus the rescaled distribution is no longer that of a thermal state.

[2]In the ideal Bose gaz regime the linear density is $n \simeq \sqrt{T/(2\pi)}g_{1/2}(e^{\mu/T})$, where $\mu < 0$ and $g_{1/2}(z) = \sum_{j=1}^\infty z^j/\sqrt{j}$ is the bose function: it fulfills $n \ll \sqrt{T}$ for $T \ll |\mu|$ and $n \gg \sqrt{T}$ for $T \gg |\mu|$

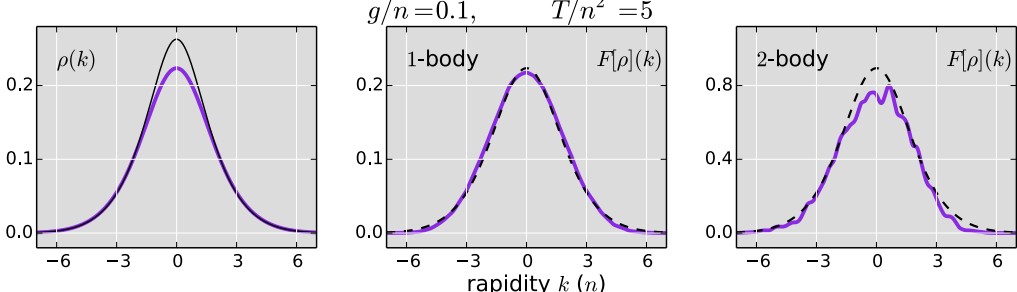

Figure 2: Left: distribution of rapidities $\rho(k)$ in the thermal state for $T = 5n^2$ and $g = 0.1n$, compared with the Bose-Eistein distribution (thin black line). Center and Right: the corresponding $F[\rho]$, calculated numerically using Eq. (7), for $K = 1, 2$ respectively. The black dashed line corresponds to $KK!\rho(k)$. The results for $K = 1$ (resp. $K = 2$) are obtained by summing over $10^4$ (resp. $2.10^3$) independent pairs of Bethe states for $N \simeq 30$ atoms. We use a Gaussian of width $\sigma = 0.15n$ to smoothen the rapidity distribution.

and vice-versa. [The fact that this is an involutive change of basis follows from the identity $\sum_{\lambda \in \frac{2\pi}{\ell}\mathbb{Z}} \frac{1}{\lambda-\mu} \frac{1}{\lambda-\mu'} = \frac{\ell^2}{4} \delta_{\mu,\mu'}$.] The Bethe states, which diagonalise the Hamiltonian, are of the form $c_{\lambda_1}^\dagger c_{\lambda_2}^\dagger \ldots c_{\lambda_N}^\dagger |0\rangle$, where $\lambda_j \in \mathbb{Z}_p$ if $N$ is odd and $\lambda_j \in \mathbb{Z}_{ap}$ if $N$ is even. Thus, while the boundary conditions are always periodic for bosons, the Hamiltonian is diagonalised on fermionic fields with either periodic or anti-periodic boundary conditions depending on the parity of atom number. The rapidities of the Bethe states are the momenta of the fermions and, on each parity sector, the Hamiltonian reduces to that of a non-interacting Fermi gas, $H = \sum_k (k^2/2) c_k^+ c_k$.

The GGE density matrix likewise separates into the parity sectors. With the projectors $\mathbf{P}_\pm = (1 \pm (-1)^N)/2$ it is of the form $\hat{\rho}_{\mathrm{GGE}} = \mathbf{P}_+ \prod_{k \in \mathbb{Z}_{ap}} \hat{\rho}_k + \mathbf{P}_- \prod_{k \in \mathbb{Z}_p} \hat{\rho}_k$, where $\hat{\rho}_k$ is diagonal in the Fock basis, so that Wick theorem applies. Using the mixed projector $\mathbf{P}_k = \mathbf{P}_+ \delta_{k \in \mathbb{Z}_{ap}} + \mathbf{P}_- \delta_{k \in \mathbb{Z}_p}$, the conserved charges may be taken as $Q_k = \mathbf{P}_k \ell c_k^+ c_k/(2\pi)$. One has $\rho(k) = \langle \mathbf{P}_k c_k^+ c_k \rangle/(2\pi)$. Expanding the commutator in Eq. (4), we obtain two terms: $\ell \langle \mathbf{P}_k \Psi^+(0) \Psi(0) c_k^+ c_k \rangle/(2\pi)$ and $-\ell \langle \Psi^+(0) \mathbf{P}_k c_k^+ c_k \Psi(0) \rangle/(2\pi)$. The first term is easily computed using $\Psi(0) = \sum_q c_q/\sqrt{\ell}$ and Wick's theorem:

$$\ell \langle \mathbf{P}_k \Psi^+(0) \Psi(0) c_k^+ c_k \rangle/(2\pi) = n\ell\rho(k) + \rho(k)(1 - 2\pi\rho(k)). \tag{12}$$

The second term amounts to computing $\langle \mathbf{P}_k c_k^+ c_k \rangle$ on a state obtained from the initial state by the removal of one atom. Crucially, after one loss the parity of the atom number has changed, which induces a sudden change of boundary conditions for the fermions. Thus, in order to use the basis that diagonalises the density matrix, in $\mathbf{P}_k c_k^+ c_k$ one must use the change of basis equation (11). Hence, we have

$$\langle \Psi^+(0) \mathbf{P}_k c_k^+ c_k \Psi(0) \rangle = \frac{4}{\ell^2} \sum_{\lambda,\lambda'} \frac{1}{\lambda-k} \frac{1}{\lambda'-k} \left\langle c^+(0) c_\lambda^+ c_{\lambda'} c(0) \right\rangle,$$

where $\lambda$ and $k$ are in different sectors. Using Wick's theorem and $c(0) = \frac{1}{\sqrt{\ell}} \sum_\lambda c_\lambda$, one gets

$$\langle \Psi^+(0) \mathbf{P}_k c_k^+ c_k \Psi(0) \rangle = \frac{4}{\ell^2} \left( n \sum_\lambda \frac{2\pi\rho(\lambda)}{(k-\lambda)^2} - \frac{1}{\ell} \left( \sum_\lambda \frac{2\pi\rho(\lambda)}{k-\lambda} \right)^2 \right).$$

Recall that the subsystem size $\ell$ is assumed to be large enough so that $\rho(k)$ varies slowly on the scale $1/\ell$. Then the sums can be replaced by integrals. In order to avoid a divergence in

$\sum_\lambda \frac{\rho(\lambda)}{(k-\lambda)^2}$, we rewrite this term as $\sum_\lambda \frac{\rho(\lambda)-\rho(k)}{(k-\lambda)^2} + \frac{\ell^2}{4}\rho(k)$. This leads to

$$\langle \Psi^+(0)\mathbf{P}_k c_k^+ c_k \Psi(0)\rangle = \frac{4}{\ell}\left(n\fint d\lambda \frac{\rho(\lambda)-\rho(k)}{(k-\lambda)^2} - \left(\fint d\lambda \frac{\rho(\lambda)}{k-\lambda}\right)^2\right) + 2\pi n\rho(k), \quad (13)$$

where '$\fint$' is the Cauchy principal value of the integral. Combining (13) with (12), we arrive at our final result for the functional $F$. The evolution of the rapidity distribution under one-body losses in the Tonks-Girardeau gas is determined by Eq. (2) with

$$F[\rho](k) = \rho(k) - 2\pi\left(\rho(k)^2 - \left(\frac{1}{\pi}\fint \frac{\rho(\lambda)d\lambda}{k-\lambda}\right)^2\right) + \frac{2n}{\pi}\fint \frac{\rho(k)-\rho(\lambda)}{(k-\lambda)^2}d\lambda. \quad (14)$$

This formula is the third main result of this paper. It shows that, in the hard-core regime, losses affect the rapidity distribution in a very non-trivial way. The functional $F$ is both non-local in rapidity space — i.e. $F[\rho](k)$ depends on $\rho(\lambda)$ for any $\lambda$, not just on $\rho(k)$ — and non-linear in $\rho(\lambda)$. This is in stark contrast with the ideal Bose gas regime, see formula (10). We stress that it is also remarkable because, even though the Tonks-Girardeau gas is mapped to a non-interacting Fermi gas, the effect of losses completely differs from the one of fermionic losses in such a gas. This is of course coming from the non-locality of the Jordan-Wigner transformation.

In Fig. 3 we compare formula (14) with numerical evaluation using the above procedure, for the rapidity distribution of a thermal state at $T = 1.02n^2$ and $g = 10^5 n$. The agreement is excellent, which further validates our numerical method.

Remarkably, we find that the evolution equation (2) with the loss term (14) can be solved analytically. For an initial distribution $\rho_0(k)$ at time $t = 0$, the distribution at time $t$ is given by the exact formula

$$\rho(k) = \text{Re}\left[\frac{\frac{i e^{-Gt}}{\pi}\int \frac{\rho_0(\lambda)d\lambda}{k-\lambda+2in_0(1-e^{-Gt})}}{1 - i2(1-e^{-Gt})\int \frac{\rho_0(\lambda)d\lambda}{k-\lambda+2in_0(1-e^{-Gt})}}\right], \quad (15)$$

where $n_0 = \int \rho_0(k)dk$ is the initial particle density and $i = \sqrt{-1}$. We defer the derivation of that formula to Appendix A.

## 7 Numerical time integration

The time evolution of the rapidity distribution is obtained by numerical time-integration of Eq. (2). Fig. 4 shows the resulting evolution of $\rho(k)$ for a gas whose initial rapidity distribution is the thermal state of Fig. 1. The time step is chosen such that the decrease of atom number is 5% of the initial atom number at each step. We perform the same calculation for a gas that lies deep into the hard-core regime, and compare to the exact result (15). The agreement is excellent, which shows that the time step is sufficiently small to provide accurate predictions. It takes about 3 days on a laptop to obtain the result shown in Fig. 4 (around 10 hours for each curve).

## 8 Inhomogeneous profiles and Generalized Hydrodynamics

Our equation (2) is readily generalized to account for the evolution of an inhomogeneous Lieb-Liniger gas, for instance in the presence of an external potential $V(x)$. At large scales

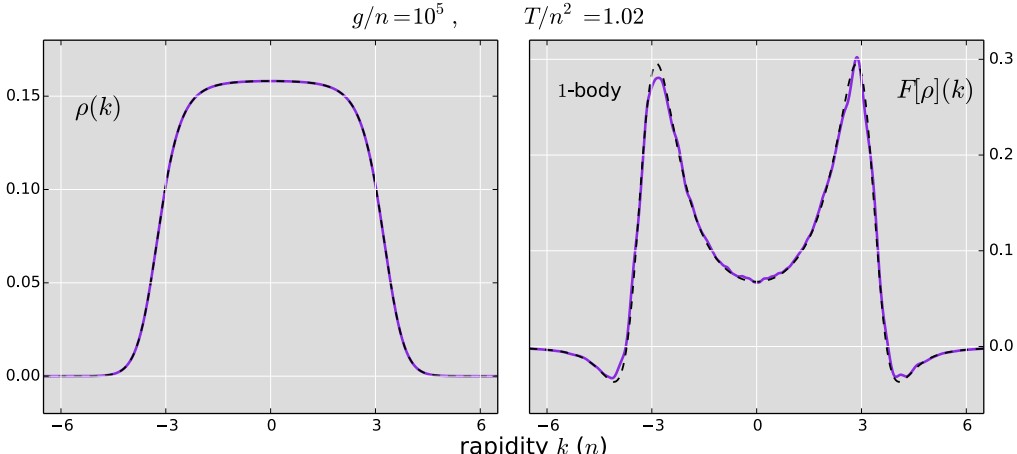

Figure 3: Left: rapidity distribution for the thermal state at $T = 1.02n^2$ and $g = 10^5 n$, compared with the Fermi-Dirac distribution (black dashed line). Right: numerical evaluation of the functional $F[\rho](k)$ for $K = 1$, compared with the exact formula (14) for the hard-core limit (black dashed line). The agreement is excellent and serves as a further validation of the numerical method. The curve is obtained by summing over $10^4$ independent pairs of Bethe states with $N \simeq 30$ particles. A Gaussian of width $\sigma = 0.12n$ is used to smoothen the rapidity distribution.

the gas is described by a position-dependent distribution of rapidities $\rho(x, k)$, which evolves according to

$$\partial_t \rho + \partial_x [v^{\text{eff}} \rho] - (\partial_x V) \partial_k \rho = -Gn^{K-1} F[\rho](k), \tag{16}$$

where the "effective velocity" $v^{\text{eff}}$ is a functional of $\rho$ defined in Refs. [59–61]. The nonzero term on the right-hand side extends the Generalized Hydrodynamics equations [60–64] to include atom losses; other types of integrability breaking situations have been studied recently [41,65–69]. Solving equation (16) is a challenging numerical problem, because it requires the calculation of $F$ for many different rapidity distributions. It may be doable with the methods we presented, using a large amount of parallelization. Analytical progress on the evaluation of the sum (7)-(8) would also be desirable and could possibly lead to drastic reduction of the computational time needed to evaluate $F$, thus facilitating a numerical solution of Eq. (16). This would lead to important improvements in the theoretical modeling of out-of-equilibrium cold atom experiments by GHD [70].

## 9  Conclusion

This work on the effect of losses in the one-dimensional Bose gas, an integrable system, calls for extensions and further studies. We can extend immediatly the results of this paper to the case, often relevant experimentally, where different loss processes occur at the same time. Then, within our assumption of slow loss process, the right-hand side of Eq.(2) is simply the sum of the contribution of each loss process. More involved further studies could explore different directions.

First, in the hard-core regime, it is possible to show from Eq. (14) that $F[\rho](k)$ generically behaves as $1/k^4$ at large $k$. Thus, in sharp contrast with its thermal equilibrium distribution, the gas typically develops $1/k^4$ tails in its rapidity distribution because of atom losses. This stems from the short-range correlations between atoms in the Lieb-Liniger model: right after a loss event, the many-body wavefunction presents a cusp at the position of the lost atom. We

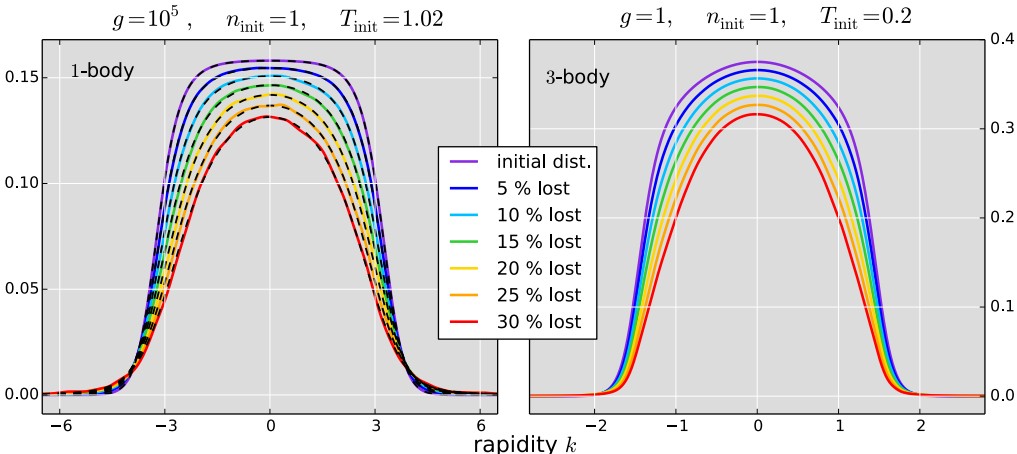

Figure 4: Evolution of the rapidity distribution under atom losses. Left: one-body losses in the hard-core regime. The initial state is the same as in Fig. 3. To benchmark our method we compare our numerical results with the analytical formula (15) (black dashed line): the agreement is excellent. Right: three-body losses, away from the hard-core regime. The initial state is the same as in Fig. 1.

expect this feature to exist also beyond the hard-core regime, and we hope to come back to this in future works. This observation is of prime importance for experimental simulations of the Lieb-Liniger model, where proper characterisation of the initial state is required. It could explain the experimental evidence for non-Gibbs ensembles in cold atoms experiment [30].

Second, it is clear that quantitative comparisons with experimental data, where the gas is usually inhomogeneous (see Eq. (16)), requires further analytical and numerical developments, in order to facilitate the evaluation of the functional $F$. Analytical progress on the thermodynamic limit of the form factors of $\Psi(0)^K$ [53] would be needed (see e.g. Refs. [71–75] for such studies of form factors of other operators), as well as methods for resumming those form factors (see e.g. Refs. [76–80]). These are very challenging tasks. A simpler problem, which might serve as a good starting point for further analytical developments, would be to compute the functional $F$ at low temperature using an effective Luttinger liquid approach, or more generally in states close to zero-entropy states where this approach can be generalized [81, 82, 82–84].

Finally, the link between the results of this paper and those previously obtained in the quasi-BEC regime, both theoretically [28, 30] and experimentally [32, 33], remains to be made.

## Acknowledgements

We thank B. Buca, B. Bertini, R. Konik and K. Kozlowski for stimulating discussions about the model, G. Masella for helpful discussions about the Monte Carlo procedure, and P. Calabrese for pointing out Ref. [53].

JD and BD thank the International Centre for Theoretical Sciences (ICTS), Bangalore, for hospitality during the program 'Thermalization, Many body localization and Hydrodynamics' (Code: ICTS/hydrodynamics2019/11).

# A  Derivation of formula (15) for one-body losses in the Tonks-Girardeau gas

In this Appendix we write the Hilbert transform of a function $f(\lambda)$ as $\mathcal{H}f(\lambda) = \frac{1}{\pi} \fint \frac{f(\mu)d\mu}{\lambda-\mu}$. We recall the following properties of the Hilbert transform:

- the complex-valued function $f(\lambda) + i\mathcal{H}f(\lambda)$ can be analytically continued to the upper half-plane: the analytic continuation is $\frac{i}{\pi} \int_{\mu \in \mathbb{R}} \frac{f(\mu)d\mu}{z-\mu}$ for $\mathrm{Im}\, z > 0$,

- applying the Hilbert transform twice, one gets $\mathcal{H}\mathcal{H}f = -f$,

- the Hilbert transform commutes with the derivative: $(\mathcal{H}f)' = \mathcal{H}(f')$. Moreover, the derivative of the Hilbert transform is minus the Hadamard finite part, which can be written as $(\mathcal{H}f)'(\lambda) = \frac{1}{\pi} \fint \frac{f(\lambda)-f(\mu)}{(\lambda-\mu)^2} d\mu$.

Using the third property, we see that our equation (2) for the evolution of the rapidity distribution with the functional $F$ given by (14) is

$$\partial_t \rho = -G\left[\rho - 2\pi\left(\rho^2 - (\mathcal{H}\rho)^2\right) + 2n(\mathcal{H}\rho)'\right]. \tag{17}$$

Here $n(t) = n_0 e^{-Gt}$ is the particle density at time $t$. Applying the Hilbert transform to both sides of that equation leads to

$$\partial_t \mathcal{H}\rho = -G\left[\mathcal{H}\rho - 4\pi\rho\mathcal{H}\rho - 2n\rho'\right]. \tag{18}$$

Here we have used the first property above: $f + i\mathcal{H}f$ is analytic in the upper half-plane, therefore $(f + i\mathcal{H}f)^2$ also is. Taking the real and imaginary part along the real axis, one gets $\mathcal{H}(f^2 - (\mathcal{H}f)^2) = 2f\mathcal{H}f$, which gives the second term in the r.h.s of (18). The third term in the r.h.s is obtained by using the second and third properties of the Hilbert transform above.

Introducing the dimensionless time $\tau = Gt$ and the analytic function in the upper half-plane

$$Q(z) = \frac{i}{\pi} \int \frac{\rho(\mu)d\mu}{z - \mu}, \quad \mathrm{Im}\, z > 0, \tag{19}$$

we see that our evolution equation becomes

$$\partial_\tau Q = -\left[Q - 2\pi Q^2 - i2n\partial_z Q\right], \tag{20}$$

which follows from adding (17) and (18). Finally, defining $Y(\tau,z) = 2\pi Q(\tau, z + 2ie^{-\tau}n_0)$, the evolution equation is simply

$$\partial_\tau Y = Y^2 - Y, \tag{21}$$

which is readily solved:

$$Y(\tau,\lambda) = \frac{Y(0,\lambda)e^{-\tau}}{1 - (1 - e^{-\tau})Y(0,\lambda)}. \tag{22}$$

Rewriting this last formula in terms of the initial rapidity distribution $\rho_0(\lambda)$, one arrives at the solution (15).

# B  Losses as evolution within a bath

The main evolution equation (3) was derived from the Lindblad equation. For pedagogical purposes, it is convenient to explain how to reproduce this equation via a toy model with

unitary Hamiltonian evolution representing the loss processes. This is done by connecting the Lieb-Liniger model to an external environment, or reservoir, able to absorb the atoms. Such toy models are in fact a standard way of deriving the full Lindblad equation, and the derivation we present follows the textbook discussions on this subject, see for instance [85, 86]. This will also make the connection of our work with recent works on perturbation of integrable models, especially [41, 65, 66, 68], more explicit. From the prespective of the toy model, Eq. (3) is nothing else but the second-order perturbation theory evolution equation discussed in these works.

Let $H$ be the Lieb-Liniger Hamiltonian, and consider $H' = H + \gamma V + H_E$ where $V$ represents the interaction with the environment, and $H_E$ is the environment's Hamiltonian. The environment interacting with the atoms at any given position, may be thought of as being composed of a family of harmonic oscillators of all frequencies. As we want to describe loss processes occurring at every point in space, the total environment is composed of one such family for every point $x$. Thus, we have canonical oscillators $c(x, \omega)$, where $x$ represents the position and $\omega$ the frequency, with

$$[c(x, \omega), c^+(x', \omega')] = \delta(x - x')\delta(\omega - \omega'). \tag{23}$$

The environment's Hamiltonian acts as

$$e^{iH_E t}c^+(x, \omega)e^{-iH_E t} = e^{i\omega t}c^+(x, \omega). \tag{24}$$

The interaction describing losses is simply that where atoms are exchanged between the Lieb-Liniger gas at position $x$, and the family of oscillators at position $x$. In order to describe exchanges which are instantaneous in time (that is, assuming that the time taken for the loss to occur is much smaller than the typical evolution timescales of the gas), the interaction – or equivalently the distribution of oscillators in the environment – is taken to be flat in frequency space, with an infinite band of frequencies. Further, in order to ensure sufficient decoherence, the frequency $\omega = 0$ is not coupled. This gives

$$V = \int_{\omega \neq 0} dx\, d\omega \left(\Psi^{+K}(x)c(x, \omega) + c^+(x, \omega)\Psi^K(x)\right). \tag{25}$$

Similarly to what is done in the main text, we make the assumptions of homogeneity of the full system, of a small interaction strength $\gamma$, and of relaxation between interaction events (which are, here, loss events). We are interested in the evolution of the Lieb-Liniger gas and the environment with respect to the full Hamiltonian $H'$ under these assumptions. In these assumptions, the relaxation is towards the GGEs with respect to the subsystem $H_0 = H + H_E$; like $H$, this subsystem is also integrable. This GGE is described by a rapidity distribution $\rho(\lambda)$ for the Lieb-Liniger gas, and a distribution of environment's oscillators $f(\omega)$ defined as $\langle c^+(x, \omega)c(x', \omega')\rangle_{[\rho, f]} = \delta(x - x')\delta(\omega - \omega')f(\omega)$. The evolution of any conserved density $q(x)$ (conserved with respect to $H_0$) under these assumptions can be obtained from a standard second-order perturbation theory, and takes the form

$$\partial_t \langle q \rangle = \gamma^2 \int_{-\infty}^{\infty} ds\, \langle [V(s), Q]v \rangle_{[\rho, f]}, \tag{26}$$

where $V(s) = e^{iH_0 s}Ve^{-iH_0 s}$ and $v = \int d\omega \left(\Psi^+(0)c(0, \omega) + c^+(0, \omega)\Psi(0)\right)$. This general equation, written, with $Q = Q_k$, as an evolution equation for the rapidity density, is at the basis of the Boltzmann kinetic formulation of perturbed integrable models [41, 65, 66, 68].

As we wish to describe loss events, and not events where particles are re-absorbed by the Lieb-Liniger gas, we take the initial state of the environment to be the vacuum, $f(\omega) = 0$. By looking at the evolution of the conserved densities $q_\omega(x) = c^+(x, \omega)c(x, \omega)$, one can show

that the environment's state stays the vacuum throughout time under the evolution equation (26). Using the vacuum property and the canonical commutation relations (23), one then finds, for all conserved densities $q_k$ of the Lieb-Liniger gas,

$$\partial_t \langle q_k \rangle = \gamma^2 \langle [\Psi(0)^{+K}, Q_k]\Psi(0)^K \rangle_{[\rho]}, \tag{27}$$

where the environment's contribution has been factored out. This is indeed Eq. (3) where $Q = Q_k$ is taken to be a conserved quantity, and where we identify $\gamma^2 = G$.

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
