# Peer review of "The effect of atom losses on the distribution of rapidities in the one-dimensional Bose gas"

_SciPost Physics, doi:SciPost Phys. 9, 044 (2020)_

## Round 1 · Referee Report · Anonymous (Referee 1) · 2020-7-23

Strengths

1: Relevance and interest of subject. 2: Exact formulation of evolution equation for rapidity distribution 3: Exact derivation of kernel (sum over form factors) 4: Exact results in two limits, one of which exhibits non trivial effects of atom losses with potential experimental consequences.

Weaknesses

1: the derivation of \rho(k) after Eqn 10 (free boson case) should be clarified.

Report

This paper aims at describing the evolution of a Bose gas (Lieb Liniger gas) taking into account atom losses, a highly relevant issue in the perspective of describing as accurately as possible the dynamics of a physical system of cold atoms. Hence the relevance and general interest of the problem. The integrable properties of the model requires the use of the full rapidity distribution to encapsulate the thermodynamical properties of the system. The authors propose an evolution equation for the rapidity distribution, characterized by a functional kernel. They then derive an exact expression for this kernel, and rexpress it in terms of sum over form factors of the Bethe states, a key object in most studies of integrable systems and measure of their observables. Numerical evaluations are proposed, and two exact resummations are identified, respectively ideal Bose gas and Tonks Girardeau limit. In both cases exact results for the the rapisity distribution are obtained, with remarkable, non trivial properties in the hard-core repulsion limit. It is undoubtedly a very interesting and very relevant paper. It is written in a rather terse and dense way, but all relevant elements are present to allow for a full (if not always easy) follow-up of the reasoning and computations.

I have some remarks and clarifications to ask for, however, which I think should be adressed to allow for final publication of this paper.

Point 1: could the authors slightly expand what they mean by "typicality of eigenstates" p4 column 2 ? and why it fails in this case ?

Point 2: I am quite confused by the derivation of the rapidity distribution after Eqn 10: -The variable \mu is not defined, to the best of my understanding; -The comment about a "rescaled" BE distribution is not clear; - I am confused by the supposedly equivalent conditions stated as " T << mu (i.e. T >> n²)" ?? there must be a misprint somewhere.

Finally there is a clear misprint in (7) where \psi(0)^K should obviously read \Psi(0)^K in the 1st line

---

## Round 1 · Referee Report · Anonymous (Referee 2) · 2020-7-30

Strengths

  1. Non-trivial quantitative predictions and exact results
  2. New method developed to solve difficult problem
  3. The topic is timely
  4. The draft reads very clearly

Weaknesses

No weaknesses

Report

In this manuscript the authors study the effects of atom losses on the quasi-stationary description for a 1D Bose gas with point-wise interactions. The topic is of manifest experimental interest, and certainly timely from the theoretical point of view. Furthermore, the draft is well-written and (despite the necessarily technical discussions) it is easy to follow.

The problem under study is difficult, because the integrability of the model makes conventional hydrodynamic descriptions not applicable, while the large number of particles make exact integrability-based calculations unfeasible. On top of that, atom losses explicitly break the unitarity of the evolution, which introduces additional complications. Despite all of this, the authors manage to provide highly non-trivial quantitative (and even analytic) predictions, which I believe is quite impressive. Finally, the method developed opens the way to further generalizations and applications to other settings.

For these reasons, I believe the draft clearly deserves publication. However, I have a few comment and questions that I list below.

  1. First, the authors do not comment on one natural numerical check which appears to be available. Namely, after Eq. (2) they show that the integral of the function F\rho must be equal to Kn(g_K). For these, there exist exact formulas in the thermodynamic limit, so that one has an analytic prediction for this number, given \rho. Thus, one could integrate numerically the functions F\rho obtained via Monte Carlo simulations, and compare against the exact thermodynamic result. Note that for K>1, this expectation value is zero in the Tonks-Girardeau limit, so this would yield a non-trivial test only for finite values of the interactions.

Did the authors perform this check? In any case, I think this would be a valuable check, and maybe the authors could comment on the numerical difference obtained, as this would give us a quantitative measure of, say, finite-size effects or other sources of numerical inaccuracy.

  1. Do the authors have an intuition regarding the double-peak shape of the function F[\rho]? Could this be expected? Is this particular of thermal states? (as compared to more generic GGEs)

  2. The authors say that the computation of F[\rho] takes~ 10 hours. On the other hand, if I understand correctly, Figure 4. requires at least 6 such subsequent computations. Does this mean that the computation time for this plot is ~60 hours? (this is just a curiosity, but the author might comment on this on the draft if they think it is relevant)

  3. In general, two- and three-body losses will happen at the same time, meaning that the Lindbladian in Eq. (1) will contain several terms, each corresponding to a different K. What would be the differences to treat this case, and how much more difficult would this be in practice?

---

## Round 2 · Author Response

We thank the referees for their work on our paper. We are glad that they appreciate this work, and we thank them for their remarks and questions. We answer their comments below.
Answer to Referee 1
Point 1: "could the authors slightly expand what they mean by "typicality of eigenstates" p4 column 2 ? and why it fails in this case ?"
"Typicality of eigenstates" is a subtle notion which has been discussed in the literature. Here we expanded that sentence (second paragraph of the second column of p.4) in order to provide at least the main aspect of what we mean by "typicality of eigenstates", and we provided additional references where more details and physical implications can be found. It now reads: "according to the idea of typicality of eigenstates –sometimes referred to as Generalized Eigenstate Thermalisation Hypothesis–, the expectation values of local observables in Bethe-Ansatz states are smooth functionals of their rapidity distribution and do not depend on details of the specific state considered (see e.g. Ref. [50-52])."
However, here, it is not the typicality of eigenstates that fails, but rather a technical trick which is based on this idea of typicality (see e.g. Ref. [53] as discussed in the text), and which has been used by experts in the field. We are simply pointing out that the trick cannot be used here because our observable is not sufficiently local, and sensitive to the rugosity of the distribution. We have slightly reorganized the end of that paragraph in order to clarify this remark. It now reads: "we find that this idea does not work in practice: the discrete nature of the rapidities induces a rugosity of their distribution at finite $\ell$, and huge system sizes $\ell$, hardly tractable numerically, would be necessary to mitigate this effect. We find that, for our purposes, the above method works and leads to reliable numerical results, while using a single Bethe state as in Ref.~[53] doesn't.
Point 2: "I am quite confused by the derivation of the rapidity distribution after Eqn 10."
We have clarified what we wanted to express there: that if the system starts in a thermal state, then as it evolves under loss, it looses its property of being thermal: it becomes non-thermal. This is expected to be true in general, but here it can be seen explicitly to be true in the ideal Bose gas regime.
We thus expanded the discussion after Eq.10 about the rapidity distribution for gases in the ideal Bose gase regime. In particular:
-
the notations $T$ and $\mu$ for temperature and chemical potential are now properly defined. We also emphasize that $\mu<0$ in the ideal Bose gas regime.
-
we expanded the discussion about the fact that the gas, initially described by a thermal state, is no longer described by a thermal state after losses occur. We also added a footnote to explain why a rescaled Bose-Einstein distribution is no longer a Bose-Einstein distribution.
-
There is no misprint concerning the condition to lie in the degenerate ideal gas regime. This regime corresponds to $\mu$ negative and large in absolute value: $|\mu| \gg T$. It corresponds to a linear density small compared to $\sqrt{1/T}$, ie small compared to the inverse of the de Broglie wavelength. We added another footnote to clarify this point.
Point 3: "there is a clear misprint in (7) where \psi(0)^K should obviously read \Psi(0)^K in the 1st line"
Thanks for catching this; we corrected that misprint.
Answer to referee 2
-
" First, the authors do not comment on one natural numerical check which appears to be available. Namely, after Eq. (2) they show that the integral of the function F\rho must be equal to Kn(g_K). For these, there exist exact formulas in the thermodynamic limit, so that one has an analytic prediction for this number, given \rho. Thus, one could integrate numerically the functions F\rho obtained via Monte Carlo simulations, and compare against the exact thermodynamic result. Note that for K>1, this expectation value is zero in the Tonks-Girardeau limit, so this would yield a non-trivial test only for finite values of the interactions.
Did the authors perform this check? In any case, I think this would be a valuable check, and maybe the authors could comment on the numerical difference obtained, as this would give us a quantitative measure of, say, finite-size effects or other sources of numerical inaccuracy."
This is a good question, however in the end it turns out the sum rule satisfied by $F$ --i.e. $\int F\rho dk = K n g_K$-- is satisfied automatically by our numerical procedure, so unfortunately it is not a useful check of its validity. The reason is the following.
We use Eq. (8) to numerically evaluate $F$. But notice that $\int \left( \sum_i \delta_\sigma (k-\lambda_i) - \sum_j \delta_\sigma (k-\mu_j) \right) dk = K$, simply because $\int \delta_{\sigma} (k-\lambda_i) dk = 1$ and the first sum has $N$ terms, while the second one has $N-K$ terms.
Therefore, integrating the right hand side of (8) with respect to $k$ simply leads to $n K \sum_{\left| { \lambda_i } \right>} p({ \lambda_i }) g_K({ \lambda_i }) ,$ where the weights $p({ \lambda_i })$ are, by definition, the ones of the GGE, see Eq. (5) in the text.
Therefore, by integrating our numerical functional $F[\rho] (k)$ w.r.t. $k$, one is simply measuring the average value of $g_K$ in the GGE (times $n K$), so it gives the expected result by construction.
[Of course, one might then worry about the accuracy of the expectation value of $g_K$ obtained from that calculation in finite size, and about how fast it approaches the thermodynamic limit as the size increases. But this is a question about the evaluation of expectation values of local observables in GGEs which has been studied elsewhere --see for instance Ref. [47]--, and is not really the goal of this paper.]
-
"Do the authors have an intuition regarding the double-peak shape of the function F[\rho]? Could this be expected? Is this particular of thermal states? (as compared to more generic GGEs)"
Unfortunately no, we do not have an intuition regarding the shape of the functional F. The larger values of F seem to be around rapidities where the distribution $\rho(\theta)$ has larger variations; but we do not have a clear intuition for why this is so at this point.
-
"The authors say that the computation of F[\rho] takes~ 10 hours. On the other hand, if I understand correctly, Figure 4. requires at least 6 such subsequent computations. Does this mean that the computation time for this plot is ~60 hours? (this is just a curiosity, but the author might comment on this on the draft if they think it is relevant)"
This is correct, it took about 3 days to get the data shown in Fig. 4. We added that comment at the end of that paragraph, as suggested.
-
"In general, two- and three-body losses will happen at the same time, meaning that the Lindbladian in Eq. (1) will contain several terms, each corresponding to a different K. What would be the differences to treat this case, and how much more difficult would this be in practice?"
Treating the case where different loss processes (like 2 and 3-body processes) occur at the same time is not difficult. Within our approximation of slow loss process, we expect the derivative of the rapidity distribution to be linear in each of the G coefficients. Thus Eq.2 acquires, on the right hand side, the sum of the contributions of each loss process computed independently. We added a sentence in the conclusion to mention this. On a practical level, the time required for the calculation taking into account two loss processes will be the sum of the times needed to compute $F$ for each loss process.

---

## Round 2 · List of Changes

List of changes
-After Eq.1, to be sure there is no confusion, we specify $\Psi=\Psi(x)$
-After Eq. 2, we added a sentence and new references: "The evolution ofrapidity distributions, or GGEs, under general Lindbla-dian dynamics has been studied in [43–46], however theparticular problem of losses has not been addressed yet."
-In the explanation that $[H,q]$ is local, we slightly change the sentence to be more clear. (This is in the paragraph "The functional F has an expectation value of a local ob-servable ...")
-After Eq.(3), to be more clear we replace "the notation $[\rho]$" by "The notation $\left< \dots \right>_{[\rho]}$"
-In Eq.7, we replaced $\psi$ by $\Psi$
-After Eq. 7, in the physical interpretation, we slighlty modify a sentence, to be more clear: we replaced "In such a case, the final rapidity distribution $q_k= (1/\ell)\sum_i \delta_\sigma (k-\mu_i) $" by "In such a case, the final value of $q_k$ is $ q_k= (1/\ell)\sum_i \delta_\sigma (k-\mu_i) $"
-In the paragraph begining with "The procedure is computationally costly...", we replaced "k=0.06" by "k=0.06n"
-We modified the paragraph begining with "Finally, we would like to stress that, in principle, for a sufficiently large system size $\ell$, a single Bethe state would be sufficient...". To be more clear, we extend a little this paragraph and added references.
-After Eq.10, we modified the paragraph discussing the fact that the system, if initially in a thermal state, do not stay in a thermal state as losses occur. Notation are explicitly given, we tried to be more clear, and footnotes are added.
-A $1/\ell$ was missing in the equation just before eq.13. We corrected this mistake.
-In the paragraph "Numerical time integration", we added the sentence "It takes about 3 days on a laptop to obtain the result shown in Fig.4 (around 10 hours for each curve)"
-In the conclusion, we added a discussion about the case where different loss processes (with different K) occur at the same time.

You are currently on this page
Jerome Dubail on 2020-09-14 [id 965]
We thank the referees for their work on our paper. We are glad they appreciate our work. We thank them for their remarks and questions. We answer their comments below.
Answer to Referee 1
Point 1: "could the authors slightly expand what they mean by "typicality of eigenstates" p4 column 2 ? and why it fails in this case ?"
"Typicality of eigenstates" is a subtle notion which has been discussed in the literature. Here we expanded that sentence (second paragraph of the second column of p.4) in order to provide at least the main aspect of what we mean by "typicality of eigenstates", and we provided additional references where more details and physical implications can be found. It now reads: "according to the idea of typicality of eigenstates –sometimes referred to as Generalized Eigenstate Thermalisation Hypothesis–, the expectation values of local observables in Bethe-Ansatz states are smooth functionals of their rapidity distribution and do not depend on details of the specific state considered (see e.g. Ref. [50-52])."
However, here, it is not the typicality of eigenstates that fails, but rather a technical trick which is based on this idea of typicality (see e.g. Ref. [53] as discussed in the text), and which has been used by experts in the field. We are simply pointing out that the trick cannot be used here because our observable is not sufficiently local, and sensitive to the rugosity of the distribution. We have slightly reorganized the end of that paragraph in order to clarify this remark. It now reads: "we find that this idea does not work in practice: the discrete nature of the rapidities induces a rugosity of their distribution at finite $\ell$, and huge system sizes $\ell$, hardly tractable numerically, would be necessary to mitigate this effect. We find that, for our purposes, the above method works and leads to reliable numerical results, while using a single Bethe state as in Ref. [53] doesn't.
Point 2: "I am quite confused by the derivation of the rapidity distribution after Eqn 10."
We have clarified what we wanted to express there: that if the system starts in a thermal state, then as it evolves under loss, it looses its property of being thermal: it becomes non-thermal. This is expected to be true in general, but here it can be seen explicitly to be true in the ideal Bose gas regime.
We thus expanded the discussion after Eq.10 about the rapidity distribution for gases in the ideal Bose gase regime. In particular:
-the notations $T$ and $\mu$ for temperature and chemical potential are now properly defined. We also emphasize that $\mu<0$ in the ideal Bose gas regime.
-we expanded the discussion about the fact that the gas, initially described by a thermal state, is no longer described by a thermal state after losses occur. We also added a footnote to explain why a rescaled Bose-Einstein distribution is no longer a Bose-Einstein distribution.
-There is no misprint concerning the condition to lie in the degenerate ideal gas regime. This regime corresponds to $\mu$ negative and large in absolute value: $|\mu| \gg T $. It corresponds to a linear density small compared to $\sqrt{1/T}$, ie small compared to the inverse of the de Broglie wavelength. We added another footnote to clarify this point.
Point 3: "there is a clear misprint in (7) where \psi(0)^K should obviously read \Psi(0)^K in the 1st line"
Thanks for catching this; we corrected that misprint.
Answer to referee 2
This is a good question, however in the end it turns out the sum rule satisfied by $F$ --i.e. $\int F \left[\rho \right] (k) dk = K n g_K$ -- is satisfied automatically by our numerical procedure, so unfortunately it is not a useful check of its validity. The reason is the following.
We use Eq. (8) to numerically evaluate $F$. But notice that $\int \left( \sum_i \delta_\sigma (k-\lambda_i) - \sum_j \delta_\sigma (k-\mu_j) \right) dk = K$, simply because $\int \delta_{\sigma} (k-\lambda_i) dk = 1$ and the first sum has $N$ terms, while the second one has $N-K$ terms.
Therefore, integrating the right hand side of (8) with respect to $k$ simply leads to
$$ n K \sum_{\left| \{ \lambda_i \} \right>} p(\{ \lambda_i \}) g_K(\{ \lambda_i \}) , $$where the weights $p({ \lambda_i })$ are, by definition, the ones of the GGE, see Eq. (5) in the text. Therefore, by integrating our numerical functional $F[\rho] (k)$ w.r.t. $k$, one is simply measuring the average value of $g_K$ in the GGE (times $n K$), so it gives the expected result by construction.
[Of course, one might then worry about the accuracy of the expectation value of $g_K$ obtained from that calculation in finite size, and about how fast it approaches the thermodynamic limit as the size increases. But this is a question about the evaluation of expectation values of local observables in GGEs which has been studied elsewhere --see for instance Ref. [47]--, and is not really the goal of this paper.]
Unfortunately no, we do not have an intuition regarding the shape of the functional F. The larger values of F seem to be around rapidities where the distribution $\rho(\theta)$ has larger variations; but we do not have a clear intuition for why this is so at this point.
This is correct, it took about 3 days to get the data shown in Fig. 4. We added that comment at the end of that paragraph, as suggested.
Treating the case where different loss processes (like 2 and 3-body processes) occur at the same time is not difficult. Within our approximation of slow loss process, we expect the derivative of the rapidity distribution to be linear in each of the G coefficients. Thus Eq.2 acquires, on the right hand side, the sum of the contributions of each loss process computed independently. We added a sentence in the conclusion to mention this. On a practical point of view, the time required for the calculation taking into account two loss processes will be the sum of the times needed to compute $F$ for each loss process.
Anonymous on 2020-09-11 [id 962]
The questions and suggestions of the referee have been suitably answered. The paper is therefore acceptable for publication as it now stands.

---

## Editorial Decision

published